# Expression of novel long noncoding RNAs defines virus-specific effector and memory CD8+ T cells

William H. Hudson [1,2], Nataliya Prokhnevska [1,2], Julia Gensheimer[1,2], Rama Akondy[1,2], Donald J. McGuire[1,2], Rafi Ahmed [1,2] & Haydn T. Kissick[2,3]

In response to viral infection, CD8+ T cells undergo expansion and differentiate into distinct classes of effector cells. After clearance of the virus, a small population of long-lived memory cells persists. Comprehensive studies have defined the protein-coding transcriptional changes associated with this process. Here we expand on this prior work by performing RNA-sequencing to identify changes in long noncoding RNA (lncRNA) expression in human and mouse CD8+ T cells responding to viral infection. We identify hundreds of unannotated lncRNAs and show that expression profiles of both known and novel lncRNAs are sufficient to define naive, effector, and memory CD8+ T cell subsets, implying that they may be involved in fate decisions during antigen-driven differentiation. Additionally, in comparing mouse and human lncRNA expression, we find that lncRNAs with conserved sequence undergo similar changes in expression in the two species, suggesting an evolutionarily conserved role for lncRNAs during CD8+ T cell differentiation.

[1] Emory Vaccine Center, Emory University School of Medicine, Atlanta, GA 30307, USA. [2] Department of Microbiology and Immunology, Emory University School of Medicine, Atlanta, GA 30307, USA. [3] Department of Urology, Emory University School of Medicine, Atlanta, GA 30307, USA. Correspondence and requests for materials should be addressed to H.T.K. (email: haydn.kissick@emory.edu)

Upon antigen exposure, naive T cells proliferate and undergo differentiation into effector T cells capable of migration to areas of inflammation and targeted killing of antigen-expressing cells. After clearance of the virus, most antigen-specific CD8[+] T cells die; however, a small proportion of memory T cells remain with the capacity to respond with greatly increased kinetics to protect the host from reinfection. The protein-coding transcriptomic changes that accompany this differentiation process have been well studied. During the effector stage, cells express many genes associated with proliferation, migration, and cytotoxicity. Upon clearance of antigen, expression of many of the genes return to a naive-like state, but levels of many key transcription factors (*Tbx21*, *Prdm1*), migration molecules (*Cd44*, *Cxcr3*), and cytokine receptors (*Il12rb*, *Il18ra*) remain elevated to allow a rapid recall to the effector state[1–3]. Importantly, many of the same genetic programs observed in mice are conserved during effector and memory T cell differentiation in human subjects[4–6]. While these previous studies provided extensive insight into the protein-coding changes associated with T cell differentiation, much less is known regarding expression of non-protein-coding transcripts during this process.

RNA-sequencing has rapidly expanded the discovery and analysis of non-protein-coding genes, which comprise much of the mammalian genome. There are as many genes encoding long noncoding RNAs (lncRNAs) as protein-coding genes, and studies have identified anywhere from 19,000 to over 60,000 unique lncRNA sequences in the mammalian genome[7,8]. Importantly, lncRNAs are often expressed in a highly tissue-specific manner, implying that their functional roles may be cell type-specific. While precise functions have been reported for only a small subset of these genes, lncRNAs have been shown to exhibit diverse mechanisms of action: lncRNAs can bind to transcription factors and modulate their activity[9–12]; the noncoding gene *Xist* is required for X chromosome inactivation[13]; and many lncRNAs interact with cellular chromatin modifying machinery to modulate gene expression[14]. Furthermore, lncRNAs are aberrantly expressed in many cancers[15] and play important roles in pluripotency, brain morphogenesis, and embryonic development[16–18]. However, the lncRNA transcriptome and its changes during antigen-driven differentiation in CD8[+] T cells are poorly defined.

Here we expand upon protein-focused transcriptional studies to identify the expression of known and novel lncRNAs in human and mouse virus-specific CD8[+] T cell subsets. By performing deep RNA-sequencing of antigen-specific CD8[+] T cells at key stages of differentiation, we discover and detect known and novel transcripts, allowing reconstruction of the CD8[+] T cell transcriptome in its entirety. Many of the hundreds of previously unannotated lncRNAs we identify here are dynamically regulated during CD8[+] T cell differentiation. Importantly, we find that human and mouse CD8[+] T cell subsets can be defined not only by their protein-coding gene expression but also by their expression patterns of known and novel lncRNA genes, implying similar regulation of transcription of protein-coding and noncoding transcripts. Finally, we identify several novel lncRNAs that are homologous, syntenous, and expressed similarly in both species, suggesting an evolutionarily conserved role for these genes.

## Results

**Mouse CD8[+] transcriptome assembly reveals unannotated genes**. During viral infection, CD8[+] T cells differentiate into many different states to eliminate the pathogen and protect the host against subsequent reinfection. During acute infection, CD8[+] terminal effector and memory precursor cells are subsets with distinct gene expression profiles and fates, with long-lived memory cells arising from the latter pool[19]. Similarly, effector and central memory cells may represent distinct populations of CD8[+] T cell memory[20,21]. We sought to examine how the transcriptome of these cell types, including noncoding transcripts and previously unannotated genes, changes during virus infection-driven differentiation.

To construct the mouse CD8[+] T cell transcriptome, we isolated virus-specific CD8[+] T cell subsets from lymphocytic choriomeningitis virus (LCMV) infected mice: CD45.1[+] LCMV-specific P14 CD8[+] T cells were transferred to congenically distinct (CD45.2[+]) C57BL/6J recipient mice (Fig. 1a). One day post-transfer, these mice were infected with LCMV Armstrong, which causes an acute, rapidly-cleared viral infection. Eight days post-infection, short-lived terminal effector (TE) P14 T cells (CD45.1[+] CD8[+] Klrg1[+] CD127[−]) and memory precursor (MP) P14 T cells (CD45.1[+] CD8[+] Klrg1[−] CD127[+]) were isolated from spleens by FACS (Fig. 1a, Supplementary Fig. 1). Forty-eight days after infection, CD127[+] memory P14 cells were isolated from recipient mice and segregated into CD62L[+] central memory (CM) and CD62L[−] effector memory (EM) cells (Fig. 1a, Supplementary Fig. 2). Separately, naive CD44[−] CD62L[+] CD8[+] T cells were isolated from the spleens of uninfected CD45.1[+] P14 mice (Supplementary Fig. 3). RNA-sequencing libraries were generated from polyadenylated RNA isolated from each of these cell populations and subjected to deep sequencing ($8.5 \times 10^8$ paired-end reads total; Fig. 1b). We then used the de novo assembly program StringTie[22] to reconstruct the transcriptome of CD8[+] T cells. We also developed an additional method of spliced transcript assembly to complement this approach, which we termed intron chain extension (ICE; see Methods, Supplementary Fig. 4).

To quantify the quality of transcripts output by these programs, we calculated a support score that reflected the number of RNA reads supporting each spliced transcript generated by de novo assembly (see Equation 1, Methods). Transcripts generated by ICE and StringTie consistently had high support scores, indicating the high quality of the novel genes returned by both (Supplementary Fig. 5a). To explore the completeness of the transcriptomes generated, we calculated the number of spliced, expressed reference genes in the *Ensembl* 84 mouse transcriptome annotation with at least one transcript isoform fully recapitulated by ICE and/or StringTie (Supplementary Fig. 5b). In general, StringTie reconstructed the spliced mouse reference genome better than our algorithm. However, combining output from ICE and StringTie provided the best recapitulation of the reference transcriptome in both mice and humans (Supplementary Fig. 5b, c). Reconstruction of the noncoding genome was less complete than the protein-coding genome in both species, perhaps indicating that use of reference genome annotations may hinder the detection of cell type-specific lncRNAs (Supplementary Fig. 6). Based on these results, we used genes from the output of both ICE and StringTie to construct the novel transcriptome reported here. To exclude misidentification of unannotated splice isoforms of known genes as novel genes, we excluded novel transcripts that overlapped with reference *Ensembl* transcripts. Finally, we performed 5′ and 3′ RACE (rapid amplification of cDNA ends) on spliced transcripts generated by our assembly as well as 3′ RACE on single-exon transcripts (Supplementary Figs 5d–g and 7). We were able to amplify more than 50% of predicted single-exon transcripts, and RACE performed on spliced transcripts of novel genes produced from this assembly consistently confirmed their presence and predicted splicing patterns. Taken together, these results demonstrate that we generated a high-quality, reference-guided de novo transcriptome of mouse CD8[+] T cells, and we sought to quantify the characteristics of and the changes in this transcriptome during the response to viral infection.

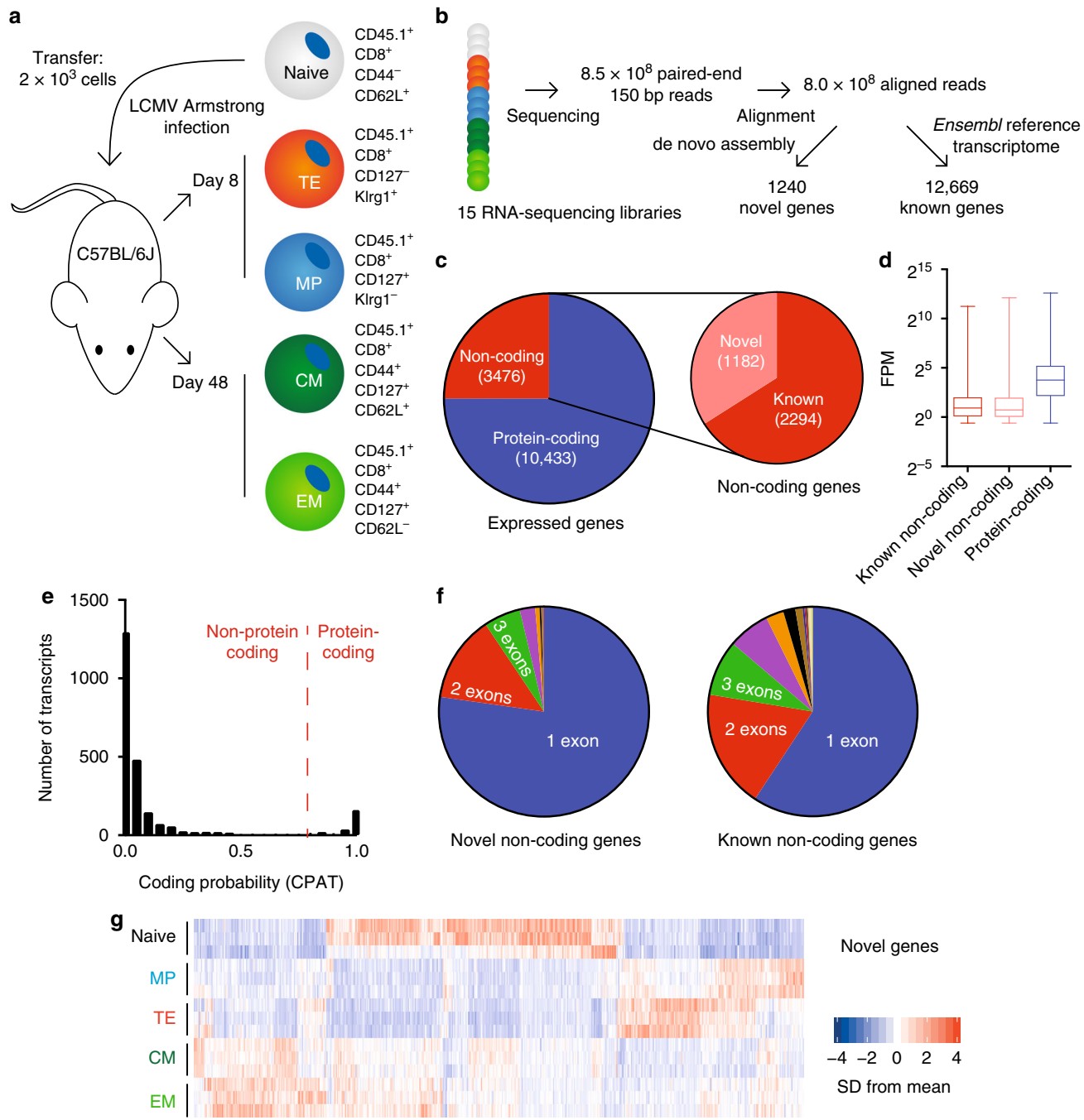

**Fig. 1** Transcriptomic analysis of antigen-specific mouse CD8+ T cells during acute infection. **a** 2000 naive CD45.1+ P14 cells were transferred to WT C57BL/6J mice that were subsequently infected with LCMV Armstrong. Eight and 48 days post-infection, indicated cell populations were isolated from splenocytes via FACS. At day 8, three pools of each cell type were isolated with three mice in each pool. At day 48, three pools of each cell type were isolated with five mice in each pool. Separately, naive cells were isolated from three uninfected CD45.1+ P14 mice. **b** From these cell populations, RNA-sequencing libraries were generated and sequenced. Resulting sequences were aligned and analyzed for the expression of both reference and novel genes. **c** Types of genes expressed in antigen-specific CD8+ T cells. **d** Average expression levels of types of genes from (**c**). **e**, **f** Protein-coding potential and exonic structure of the 1291 novel genes detected by de novo transcriptome assembly. **g** Heatmap of differentially expressed novel gene expression across all 15 samples. FPM, fragments per million. In panel (**d**), bars indicate minimum and maximum values and box indicates 25th percentile, median, and 75th percentile. FPM, fragments per million; SD, standard deviation

## Dynamic expression of lncRNAs during T cell differentiation.

During the course of LCMV Armstrong infection, we detected the expression of 13,909 unique genes among all CD8+ T cell subsets. Seventy-five percent of these genes were protein-coding genes, and the remaining 25% were noncoding (Fig. 1c). Of the noncoding genes, one-third was not present in the *Ensembl* 84 reference transcriptome annotation and thus designated as novel

genes. These novel noncoding genes had slightly higher expression levels than annotated noncoding genes; expression of both types of genes was notably lower than protein-coding genes (Fig. 1d). Of all unannotated genes, the vast majority were predicted to be noncoding by the coding potential assessment tool (CPAT), a logistic regression model that predicts coding probability from nucleotide sequences[23] (Fig. 1e). Most of the novel

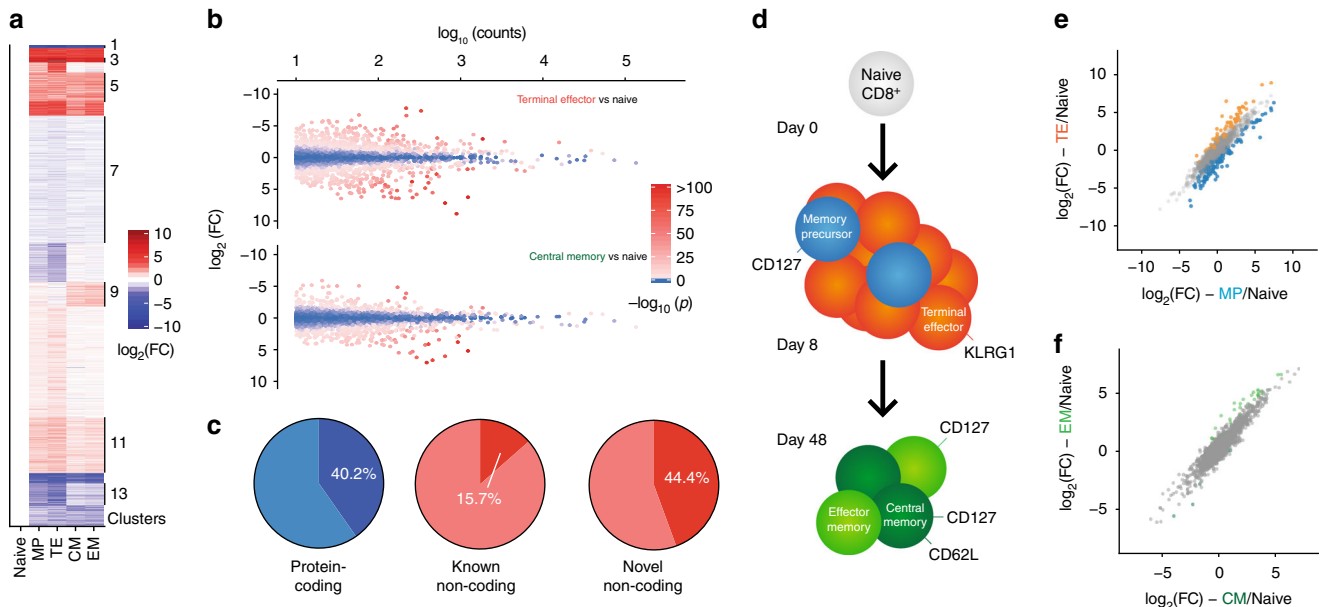

**Fig. 2** Mouse CD8+ T cell differentiation is marked by large changes in lncRNA gene expression. **a** Heatmap showing expression of all differentially expressed genes in CD8+ T cells during the response to LCMV Armstrong. **b** MA plot of noncoding genes in day 8 terminal effector (TE) and day 48 central memory (CM) CD8+ T cells compared to naive cells. **c** 40.2% of expressed, known protein-coding genes are differentially expressed throughout CD8+ T cell differentiation, compared to 15.7% of previously-annotated lncRNA genes. However, 44.4% of lncRNA genes discovered in this study are differentially expressed throughout CD8+ T cell differentiation. **d** At day 8 and 48, two subsets of CD8+ T cells were sorted. Differences in noncoding gene expression between these subsets are shown in (**e**) and (**f**). Colored points indicate significant expression differences between the two cell subsets. FC, fold change

transcripts were comprised of a single exon, with less than 25% containing two or more exons (Fig. 1f). Despite the lack of protein-coding potential, many of these novel genes were expressed at specific points during CD8+ T cell differentiation (Fig. 1g), including genes that were expressed specifically in memory cells.

Of the 13,909 total genes expressed in mouse CD8+ T cells throughout viral infection, 5266 were differentially expressed in at least one post-infection subset compared to naive cells (Fig. 2a). In total, 884 lncRNAs were differentially expressed during infection (Fig. 2b), of which 525 were previously unannotated. Overall, 40.2% of protein-coding genes expressed throughout CD8+ T cell differentiation were differentially expressed. Strikingly, 44.4% of novel noncoding RNA genes were differentially expressed, compared to only 15.7% of previously annotated lncRNAs (Fig. 2c). Thus, our de novo transcriptome assembly identified lncRNAs that are dynamically expressed during the CD8+ T cell response to viral infection.

At day 8 post-infection, 692 noncoding genes were differentially expressed in TE CD8+ T cells compared to naive CD8+ T cells. Of these, 406 genes were discovered by our de novo transcriptome assembly and were thus previously unannotated. 181 noncoding genes were differentially expressed between TE and MP cells at day 8 (Fig. 2e), 123 of which were previously unannotated. Forty-eight days post-infection, 378 noncoding genes (230 novel) were significantly different from their expression levels in naive CD8+ T cells. However, a relatively small number of genes were differentially expressed between CM and EM cells (Fig. 2f): 167 genes in total were significantly different between the two subsets, of which 29 were noncoding (20 novel).

**lncRNA expression profiles define CD8+ T cell subsets**. Given the large numbers of lncRNA genes differentially expressed

during CD8+ T cell differentiation, we were interested in how lncRNA expression differed among previously described subsets of antigen-specific CD8+ T cells. Principal component analysis (PCA) on the expression of protein-coding, noncoding, and novel genes showed that these classes of transcripts can specifically define CD8+ T cell subsets (Fig. 3a–c). As expected, PCA performed on protein-coding gene expression (Fig. 3a) placed our 15 RNA-sequencing samples into three distinct groups: those isolated from naive cells, post-infection day 8 cells, and memory cells. Notably, day 8 MP cells were slightly biased toward the memory cluster. Strikingly, PCA using only noncoding genes yielded identical grouping of the samples, indicating that naive, effector, and memory CD8+ cells can be defined by their lncRNA expression (Fig. 3b). Similarly, when only novel genes identified by our de novo transcriptome assembly were used for PCA, naive, effector, and memory cells were once again grouped virtually identically to the protein-coding gene-based analysis (Fig. 3c). Finally, using only single- or multi-exonic lncRNA genes or novel or annotated lncRNAs alone recapitulated these sample groups (Supplementary Fig. 8), indicating that the novel genes reported here may be important for T cell differentiation and function.

To identify the expression patterns of individual lncRNAs that underlie this result, we performed affinity propagation clustering[24] to group differentially expressed genes with similar expression patterns (Fig. 2a, Supplementary Fig. 9, Supplementary Data 1). This analysis identified 14 clusters of differentially expressed genes with varied expression patterns in mouse CD8+ T cells. Noncoding genes did not differentially cluster from protein-coding genes, indicating that similar processes regulate the expression of both coding and noncoding genes (Supplementary Fig. 9b). Multiple patterns of expression were evident; for example, some noncoding genes were upregulated or repressed upon CD8+ T cell activation and remained at effector levels through the memory phase, similar to the protein-coding genes *Cd44* and *Cd200* (Fig. 3d, clusters 2 and 12). A more common

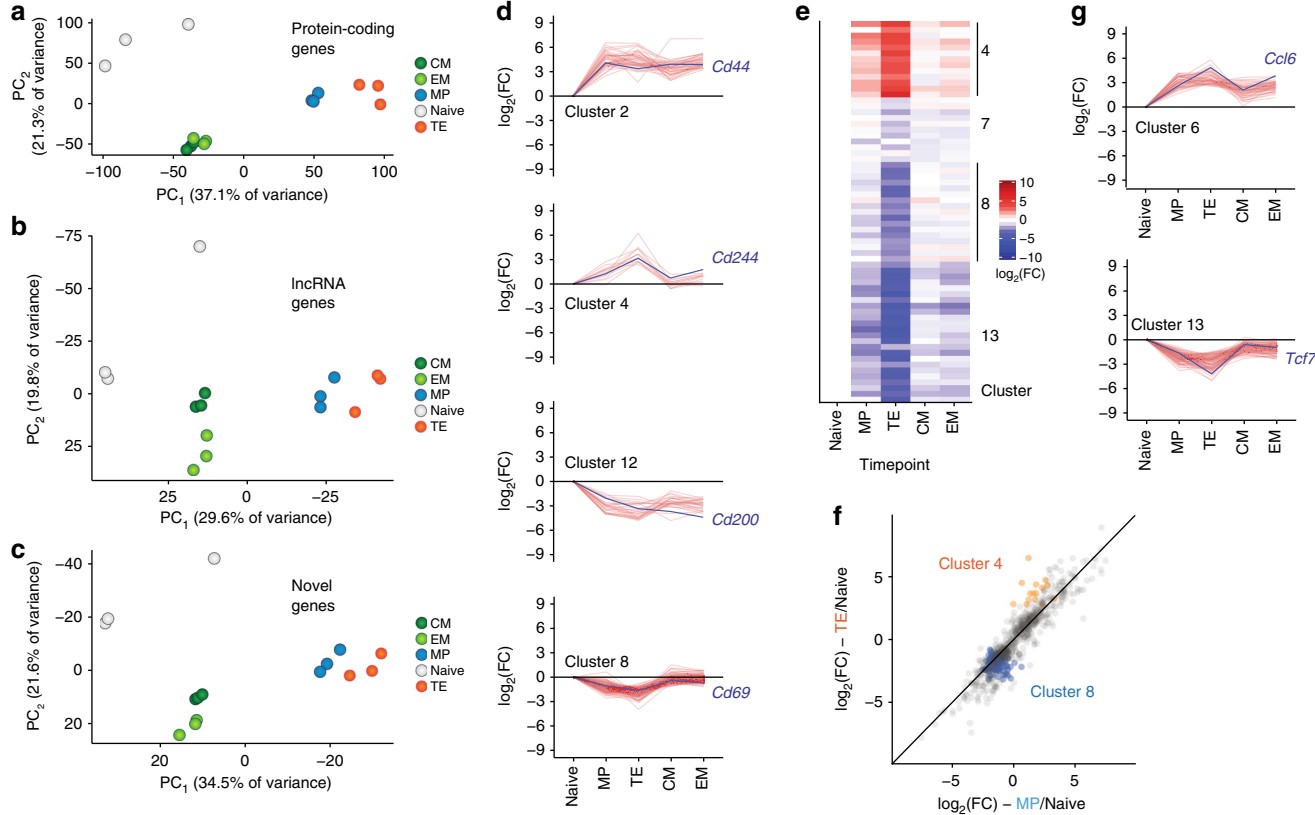

**Fig. 3** Unique lncRNA expression patterns define antigen-specific mouse CD8[+] T cell subsets. **a** Principal component analysis (PCA) of all 15 LCMV-specific CD8[+] T cell samples, using only protein-coding genes. **b** PCA of all 15 LCMV-specific CD8[+] T cell subsets, using only noncoding genes, recapitulates the PCA of protein-coding genes. **c** Expression of novel lncRNAs is likewise sufficient to discriminate among CD8[+] T cell subsets. **d** Unsupervised clustering revealed distinct expression patterns of lncRNA genes; lncRNA genes (red) and an exemplar protein-coding gene (blue) are shown for selected clusters. **e** Heatmap of noncoding genes from selected clusters that show large differences in expression between MP and TE cells at day 8. **f** log-transformed fold change (compared to naive) is shown for noncoding genes expressed in MP and TE cells. Clusters 4 and 8—which contain genes that discriminate between TE and MP cells—are shown with colored circles. **g** Clusters 6 and 13 contain genes differentially expressed between MP and TE cells

pattern was increased or decreased transcription at the effector state with a return to naive levels in the memory phase (Fig. 3d, clusters 8 and 4). Genes returning to their naive expression level in the memory phase also explains the intermediate coordinates of memory cells on the first principal component of both protein-coding and novel genes (Fig. 3a–c). lncRNA genes with differential expression between day 8 MP and TE cells were present in several clusters (clusters 4, 8, and 13 for example; Fig. 3e–g), suggesting that lncRNAs may be involved in fate decisions early in T cell differentiation. The expression of genes within these clusters generally differed from naive cells in both MP and TE cells, with the two populations being distinguished by the magnitude of that change (Fig. 3e–g).

**lncRNA expression in human antigen-specific CD8[+] T cells.** Together, the above results indicate that lncRNAs are expressed at specific stages of CD8[+] T cell differentiation in response to viral infection and that lncRNA expression is sufficient to define specific CD8[+] T cell subsets. To extend our findings to human virus-specific CD8[+] T cells, we analyzed RNA-sequencing data of tetramer-sorted CD8[+] T cells from healthy human volunteers given the live attenuated YFV-17D yellow fever vaccine (Fig. 4a)[5]. Like LCMV Armstrong infection in mice, this vaccine causes an acute, transient viremia with generation of a robust effector and memory CD8[+] T cell response[25]. Fourteen and 28 days following yellow fever vaccination, bulk A2-NS4B[214]-specific CD8[+] T cells

were sorted from PBMCs (Fig. 4a). To obtain yellow fever-specific memory cells, bulk tetramer-positive CD8[+] T cells were sorted from donors who had received the yellow fever vaccine at least three years prior to sample collection. From these sorted cells, RNA was isolated, sequencing libraries prepared, and deep sequencing performed to reconstruct the human antigen-specific CD8[+] T cell transcriptome (Fig. 4b).

After performing an identical workflow as above for de novo transcriptome assembly and differential expression analysis on this data, we identified 15,507 genes expressed during human CD8[+] T cell differentiation. In total, 3165 of these genes were noncoding transcripts that had previously been annotated. In total, 1098 additional undescribed genes from were also discovered using this process (Fig. 4c). Similar to our results from mouse LCMV infection, protein-coding genes were, on average, more highly expressed than noncoding genes (Fig. 4d), regardless of whether these genes were from the reference transcriptome or identified by de novo assembly. Likewise, most identified novel transcripts were monoexonic and displayed a lower protein-coding potential than genes identified during mouse de novo transcriptome assembly (Fig. 4e, f). Like mouse CD8[+] T cell differentiation, noncoding genes identified by de novo assembly exhibited stage-specific expression patterns (Fig. 4g). In sum, these data reveal, as in mice, hundreds of previously undescribed transcripts in CD8[+] T cells, many of which are expressed only during specific states of T cell differentiation.

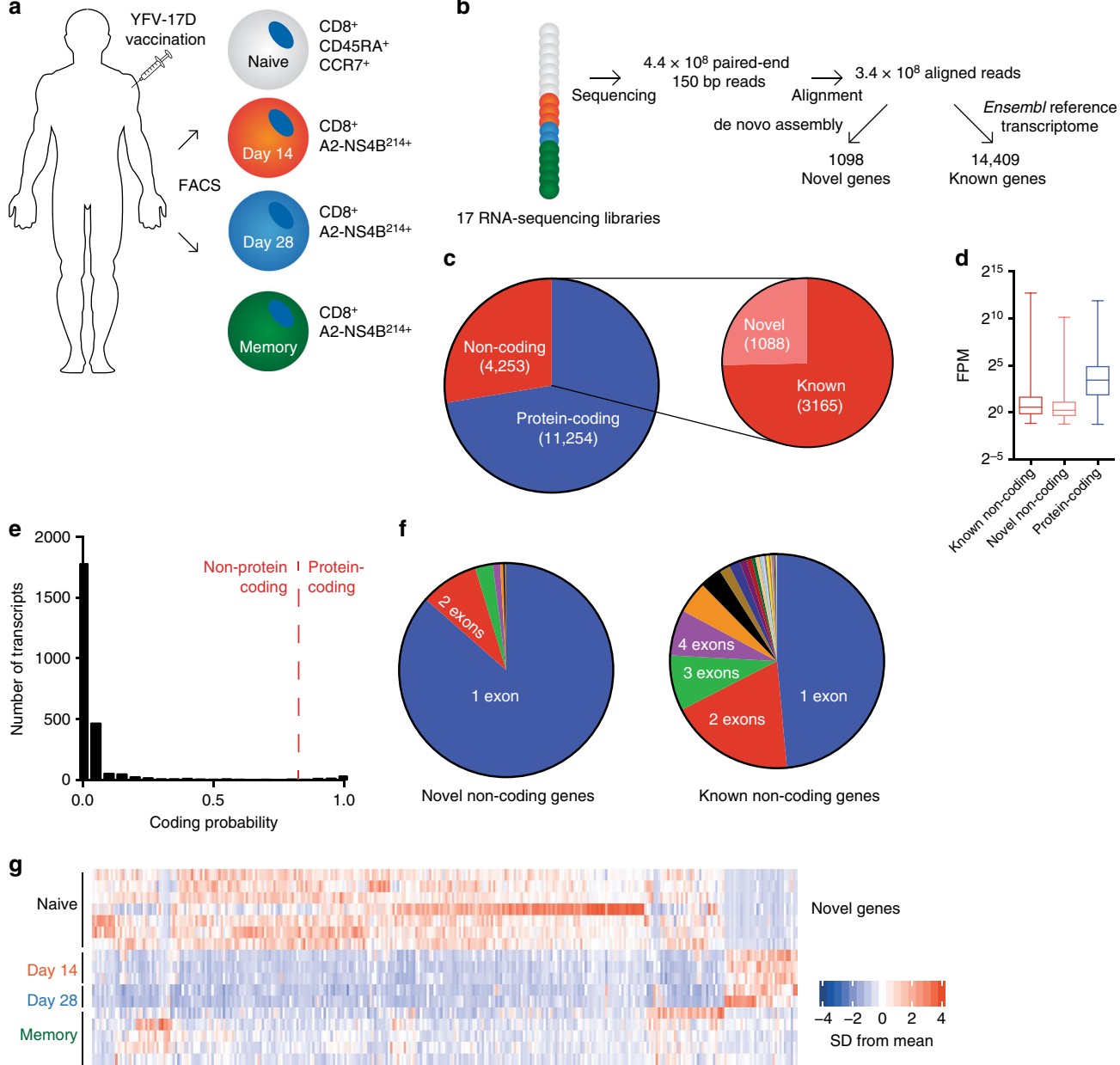

**Fig. 4** Transcriptomic analysis of antigen-specific human CD8$^+$ T cells during acute infection. **a** Antigen-specific CD8$^+$ T cells were isolated from healthy volunteers at indicated time points following administration of the live attenuated yellow fever-17D (YF-17D) vaccine[5]. **b** From these cell populations, RNA-sequencing libraries were generated and sequenced. Resulting sequences were aligned and analyzed for the expression of both reference and novel genes. **c** Types of genes expressed in human antigen-specific CD8$^+$ T cells. **d** Average expression levels of types of genes from (**c**). **e**, **f** Protein-coding potential and exonic structure of the novel genes detected by de novo transcriptome assembly. **g** Expression of differentially expressed novel genes across all samples. In panel (**d**), bars indicate minimum and maximum values and box indicates 25th percentile, median, and 75th percentile. FPM, fragments per million; SD, standard deviation

After differential expression analysis, we found that expression of 4099 genes was significantly changed during at least one time point during human CD8$^+$ T cell differentiation (Fig. 5a), 838 of which were noncoding (Fig. 5b). 28.3% of expressed protein-coding genes significantly changed over the course of infection (Fig. 5c). Of noncoding genes, novel genes were again more likely to be differentially expressed: 28.3% of these genes were significantly changed over the course of infection, compared to 16.7% of previously annotated noncoding genes. Changes in gene expression between antigen-specific CD8$^+$ T cells between day 14 and 28 post-infection were minimal: 144 genes differed between these time points (Fig. 5d, e).

To determine the role of lncRNAs in defining the transcriptional profiles of human antigen-specific CD8$^+$ T cells, we performed PCA on human yellow fever-specific cells using the expression levels of protein-coding, noncoding, and novel genes. Similar to the results found in the mouse LCMV model (Fig. 3a–c), naive, effector, and memory CD8$^+$ T cells formed three distinct clusters in all principal component analyses (Fig. 6a–c). Post-infection day 14 and 28 cells were not readily differentiated by PCA, as expected by the use of the bulk antigen-specific population. Once again, PCA performed on both known noncoding genes and the novel noncoding genes identified here yielded similar grouping as the exclusive use of protein-coding

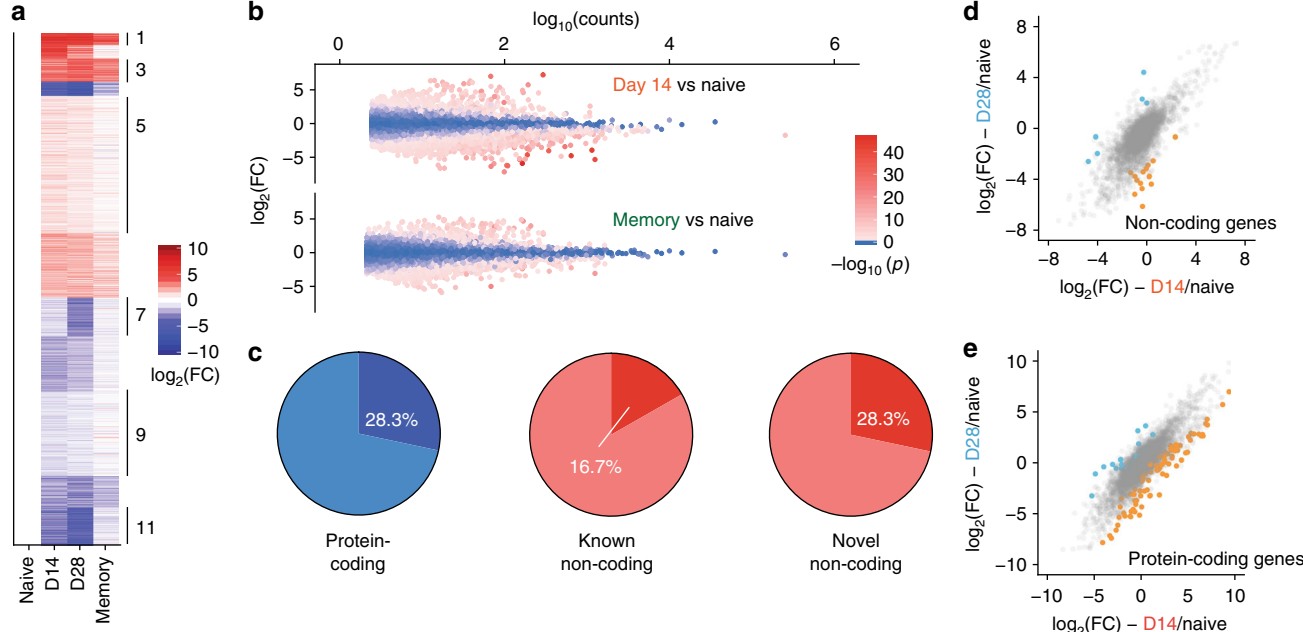

**Fig. 5** Human CD8+ T cell differentiation is marked by large changes in lncRNA gene expression. **a** Heatmap showing expression of all differentially expressed genes in human CD8+ T cells during the response to yellow fever vaccination. **b** MA plot of noncoding genes in day 14 and memory yellow fever virus-specific CD8+ T cells, compared to naive. **c** In humans, 28.3% of expressed protein-coding genes are differentially expressed throughout CD8+ T cell differentiation, compared to 16.7% of previously annotated lncRNA genes. In line with results from mice, a higher proportion of novel lncRNA genes (28.3%) are differentially expressed during CD8+ T cell differentiation compared to known lncRNA genes. **d**, **e** Changes in noncoding gene expression between day 14 and day 28 antigen-specific CD8+ T cells. Colored points indicate significant expression differences between the two cell subsets. FC, fold change

genes. To determine patterns of coding and noncoding gene expression underlying these results, we again used affinity propagation clustering, which yielded 11 clusters of gene expression (Fig. 5a, Supplementary Fig. 10, Supplementary Data 2). Similar to the mouse LCMV model, protein-coding and noncoding genes did not separate into distinct clusters (Supplementary Fig. 10). Comparable gene expression patterns also pervaded human lncRNA (and protein-coding) gene expression (Fig. 6d), such as genes that were upregulated upon activation and remained elevated after viral clearance (Cluster 3), or returned to naive (Cluster 2) or near-naive levels (Cluster 1). Many downregulated genes also returned to naive levels after memory formation (Cluster 11), but some genes remained at levels lower than naive CD8+ T cells (Cluster 10). Taken together with the data from the mouse LCMV model, these results indicate that noncoding genes contribute considerably to the unique identities of naive, effector, and memory antigen-specific CD8+ T cell subsets in multiple mammalian species.

**Similar expression of conserved lncRNAs in CD8+ T cells**. The two data sets described here provide an exceptional resource for the conservation of lncRNA gene expression, an intensely studied topic[26–28], across two mammalian species during an identical physiological process: the CD8+ T cell response to viral infection.

To determine the extent of cross-species sequence similarity between lncRNA transcripts expressed in murine and human T cells, we used BLAST to compare repeat-masked mouse lncRNA transcripts to the human transcriptome and vice-versa. Only a small fraction of mouse lncRNA genes contained homologous sequence to a human transcript, and this was not dependent on the prior annotation status of the gene (Fig. 7a). Of the mouse lncRNA genes with a homologous human transcript, sequence similarity was typically found with human protein-

coding genes and pseudogenes (Fig. 7b). A similar pattern was found with human lncRNA genes with a homologous mouse transcript (Fig. 7c). Many lncRNA genes with shared sequence had multiple homologous loci throughout the opposite species' genome (Fig. 7d, e). For example, one novel gene expressed in murine T cells on chromosome 10 (chr10-gene-81985877) shared homology with over 200 human transcripts. The matches to this novel transcript were almost entirely *Gapdh* pseudogenes, of which there are at least 62 in humans and 331 in mice[29]. Transcripts such as these are unannotated pseudogenes that have been copied many times in the genome, and may not represent novel functional sequences.

In addition to sequence conservation, shared synteny (that is, conserved chromosomal location of genes between species) has been proposed as a mechanism for identifying evolutionarily conserved lncRNAs with potential functional importance[26,30]. Thus, we hypothesized that mouse lncRNAs and human genes with which they share sequence homology and synteny may be regulated similarly during CD8+ T cell differentiation in both species. To identify these genes, we identified neighboring genes in both species of human-mouse gene pairs identified by our BLAST analysis above and selected those pairs with identical neighboring genes as having shared synteny. We then compared the fold change of these genes in mice and humans at the effector (Fig. 7f) and memory time points (Fig. 7g). Strikingly, no correlation of expression was found at either time point for genes that shared sequence but not synteny ($p = 0.91$ and 0.36, respectively). However, genes with shared synteny and sequence showed a moderate correlation of expression at both the effector and memory time points ($r = 0.36$ and 0.46, $p = 0.0007$ and 0.0008, respectively).

To compare the correlation of homologous lncRNA expression between human and mouse CD8+ T cells with that of protein-coding homologs, we used the *Ensembl*/biomaRt database to

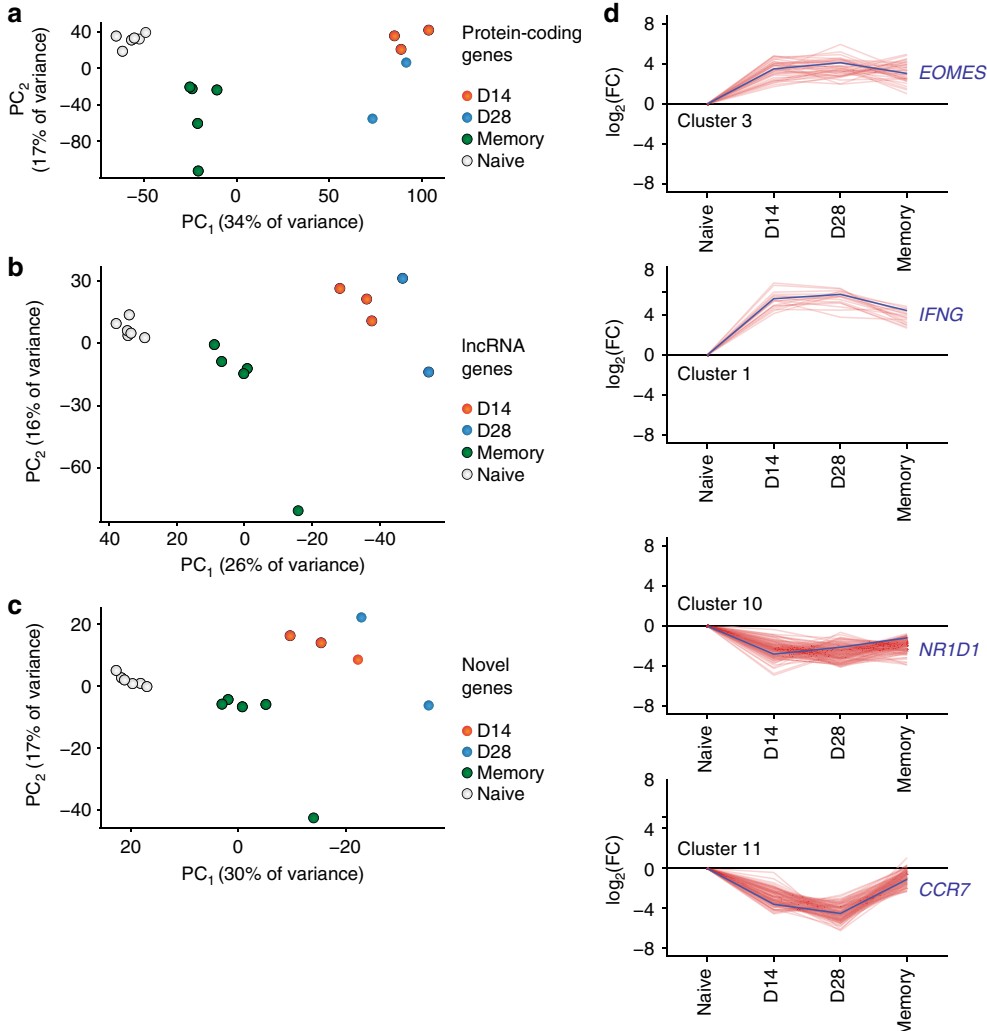

**Fig. 6** Unique lncRNA expression patterns define time points of human CD8$^+$ T cell differentiation. **a** Principal component analysis (PCA) of all human yellow fever-specific CD8$^+$ T cell samples, using only protein-coding genes. **b** PCA of human yellow fever-specific CD8$^+$ T cell subsets, using only noncoding genes, recapitulates the PCA of protein-coding genes. **c** Likewise, expression of novel lncRNAs is sufficient to discriminate CD8$^+$ T cell subsets. **d** Unsupervised clustering revealed distinct expression patterns of lncRNA genes; lncRNA genes (red) and an exemplar protein-coding gene (blue) are shown for selected clusters

identify human/mouse ortholog pairs of protein-coding genes. Naive CD8$^+$ T cells from mice and humans expressed a similar repertoire of protein-coding genes (Supplementary Fig. 11a), and these genes exhibited statistically significant correlation upon activation and differentiation to effector or memory cells (Supplementary Fig. 11b, c). However, the coefficient of correlation was modest: $r = 0.32$ and $0.22$ at the effector and memory time points, respectively. Thus, known mouse/human protein-coding ortholog pairs exhibit similar correlation of expression to the syntenic lncRNA pairs described above.

## Discussion

In this study, we aimed to define the expressed polyadenylated lncRNA repertoire of virus-specific CD8$^+$ T cells and define the changes that occur during activation and subsequent generation of immune memory. To that end, we generated a high-confidence transcriptome of antigen-specific CD8$^+$ T cells and discovered over 1000 previously unannotated transcripts in both mouse and human CD8$^+$ T cells. Approximately one-quarter of genes expressed in antigen-specific CD8$^+$ T cells in both mice and humans were noncoding, indicating that lncRNAs are a

significant proportion of the transcriptional output of T cells in their various differentiation states.

Importantly, the lncRNA expression profile of CD8$^+$ T cells was sufficient to identify the distinct naive, effector and memory populations. The striking separation of CD8$^+$ T cell subsets by lncRNA expression corroborates and extends previous studies showing that lncRNA expression is tissue specific[26,31]. Indeed, we demonstrate that lncRNA expression may be specific to particular differentiation states of a given cell type. Analyses of lncRNAs from cancer samples show that few lncRNAs are expressed across all tissues[15], suggesting that many of the novel lncRNAs we define here may be specific to T cells. Additionally, our results highlight the importance of sequencing a given cell type throughout its differentiation process in order to identify its full transcriptional repertoire.

These results also have implications for CD8$^+$ T cell function itself. For example, both protein-coding and noncoding genes distinguish terminal effector and memory precursor cells at day 8 after LCMV infection in mice (Fig. 3). This suggests that the differentially expressed lncRNAs at this time point may be involved in the cell fate decisions of effector cells. Additionally, since protein-coding and noncoding genes share similar

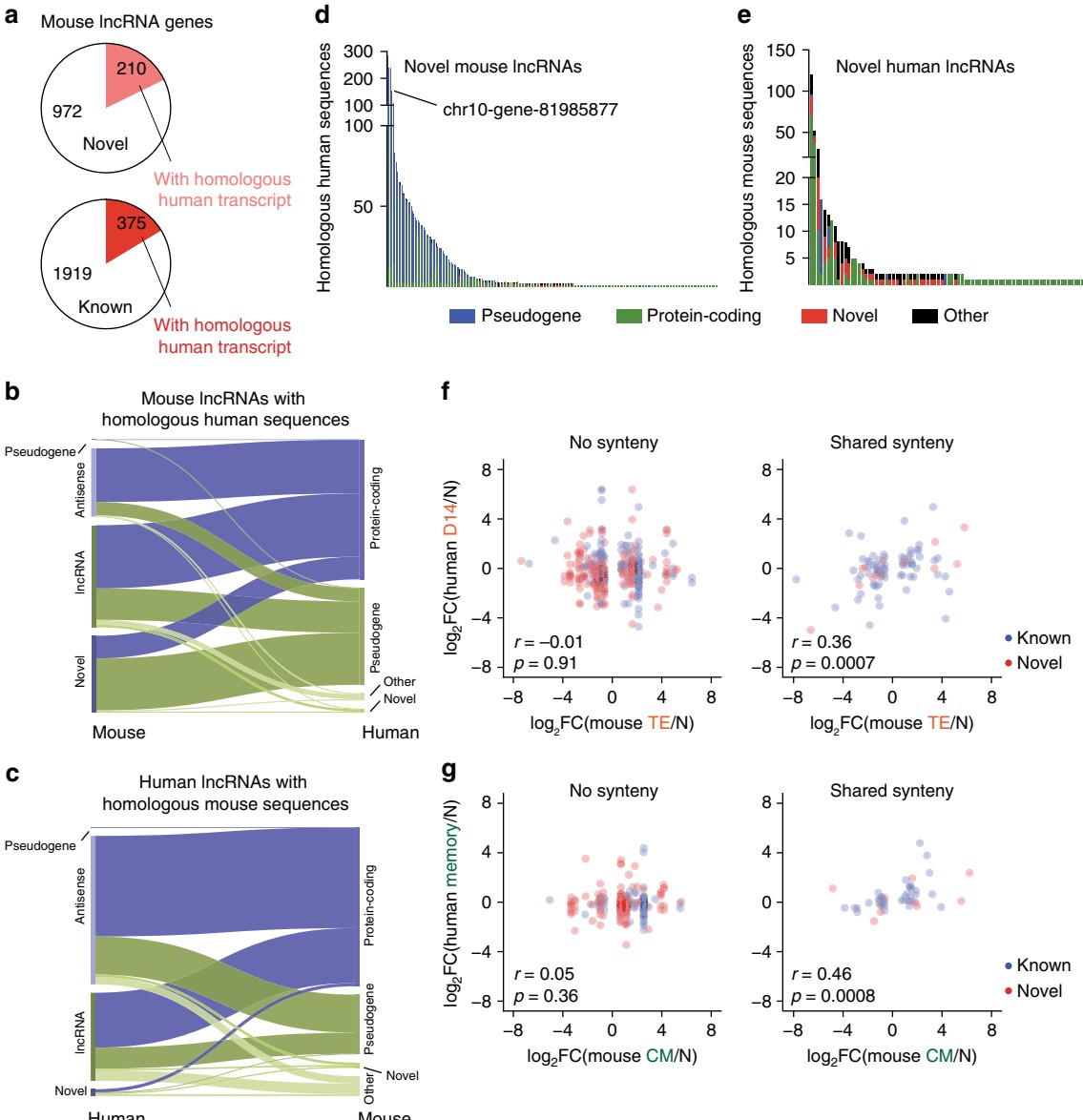

**Fig. 7** lncRNAs with conserved sequence and shared synteny exhibit similar expression patterns in human and mouse CD8$^+$ T cell differentiation. **a** Number of novel and previously annotated mouse lncRNA genes with detectable sequence homology to a human transcript. **b**, **c** Composition of noncoding RNA with any homolog in the opposite species. **d** Number and biotype of homologs of novel mouse genes with significant sequence similarity to human genes (210). **e** Number and biotype of homologs of novel human genes with significant sequence similarity to mouse genes (96). **f**, **g** Correlation of expression changes in homologous lncRNAs with or without shared synteny. In panels (**f**) and (**g**), Pearson's *r* is shown, with *p*-value calculated using a *t* distribution

expression patterns in CD8$^+$ T cells (Supplementary Figs 9, 10), it is likely that lncRNAs are regulated by similar transcription factors as their protein-coding counterparts, such as Blimp-1 or T-bet in the case of memory differentiation[32].

Given the possible important role of lncRNAs in T cell differentiation described above, we investigated the correlation of expression of lncRNAs in human and mouse CD8$^+$ T cell differentiation. Many of the novel genes we identified were highly homologous to known protein-coding genes, suggesting that these transcripts are the result of duplication events and have lost their protein-coding potential due to mutation or rearrangement. In some cases, we found examples of novel lncRNAs that had hundreds of homologs throughout the genome (Fig. 7d, e), which are unlikely to represent novel functional sequences. However, some of these genes are antisense transcripts, which could have potential regulatory roles against their homologous genes[33,34].

In addition to the highly duplicated transcripts, a small subset of homologous genes had a syntenic copy in both the mouse and human genomes; strikingly, expression of these gene pairs was correlated during CD8$^+$ T cell differentiation in humans and mice (Fig. 7f, g). In fact, these gene pairs showed a comparable degree of expression correlation to protein-coding homolog pairs (Supplementary Fig. 11). Compared to lncRNA genes that either had no homologous sequence in the converse species or did not exhibit shared synteny, this lncRNA class represents a small portion of expressed genes. A previous genome wide cross-species study showed that the vast majority of human lncRNA families are primate-specific; however, ancient (>90 Mya) lncRNAs showed more conserved patterns of expression and higher conservation of promoter transcription factor binding sites[26]. This finding is consistent with our data, and defines the syntenic pairs described here as excellent targets for future functional studies.

In conclusion, we identified long noncoding RNA as an important transcriptional output of CD8$^+$ T cells. Expression of lncRNA genes is regulated during CD8$^+$ T cell differentiation, with lncRNA expression sufficient to define T cell subsets previously characterized by expression of protein-coding genes.

## Methods

**LCMV infection and cell sorting**. For collection of LCMV Armstrong-infected mouse samples, $2 \times 10^3$ P14 cells from a single seven-week-old CD45.1$^+$CD45.2$^-$ P14 (LCMV-gp33 specific transgenic TCR) donor mouse (C57BL/6J background) were injected intravenously into 24 seven-week-old C57BL/6J mice (Jackson Laboratories, Bar Harbor, ME). The next day, these mice were injected intraperitoneally with $2 \times 10^5$ pfu of LCMV Armstrong. At day 8, effector cells were isolated from three groups of spleens (three mice per group) using FACS (Fig. 1, Supplementary Fig. 1). Memory cells were collected by FACS from CD8$^+$ T cell enriched splenocytes (using the Stemcell Mouse CD8$^+$ T cell isolation kit) 48 days after infection from three groups of five mice each (Fig. 1, Supplementary Fig. 2). Naive cells were separately collected from the spleens of three LCMV-naive CD45.1$^+$CD45.2$^-$ P14 mice (C57BL/6J background; Supplementary Fig. 3).

Antibodies (clones; supplier) used for flow sorting were CD4 (GK1.5; Biolegend); CD19 (6D5; Biolegend); CD45.1 (A20; Biolegend); CD8 (53-6.7; Biolegend); CD127 (SB/199; Biolegend); KLRG1 (2F1; Southern BioTech); CD62L (MEL-14; Thermo Fisher); and CD44 (IM7; Biolegend). All antibodies were used at a dilution of 1:100. Staining was performed by incubating indicated antibodies with cells at a concentration of $4 \times 10^7$ cells ml$^{-1}$ in staining buffer (PBS with 2% FBS and 2 mM EDTA) for 30 min on ice, followed by three washes with staining buffer.

Animal experiments were approved by and performed in accordance with guidelines from the Institutional Animal Care and Use Committee of Emory University.

**RNA Isolation and sequencing**. RNA was isolated from sorted cells of LCMV-infected mice with the Qiagen AllPrep Micro Kit. Library preparation and sequencing were performed by the HudsonAlpha Genomic Services Laboratory via their low-input RNA-sequencing protocol: poly(A)-selected RNA was amplified using the Ovation RNA-Seq System V2 kit (Nugen) and libraries were sequenced on a HiSeq 2500 instrument ($2 \times 125$ nucleotide reads). Average insert size across all samples was 199 bp ± 97 (mean ± s.d.).

Human RNA-sequencing data were previously reported[5] and processed identically to the mouse data reported here.

**Rapid amplification of cDNA ends (RACE)**. RNA was amplified from C57BL/6J mice as described above. cDNA amplification[35] and RACE[36] were performed as described previously. Briefly, splenic CD8$^+$ T cells from uninfected mice were isolated using the Stemcell Mouse CD8$^+$ T cell isolation kit. RNA was isolated with the Qiagen RNeasy mini kit, and cDNAs amplified via polymerase chain reaction using poly(dT) and template-switching primers[35]. This amplified cDNA was used as template for RACE reactions[36]. For RACE, primers were designed as shown in Supplementary Figs 5 and 7.

**De novo transcriptome assembly**. Cutadapt was used to remove adapter sequences and to quality trim reads[37], and trimmed reads were aligned to the reference genome using the STAR aligner[38]. The resulting alignments were subjected to the genome-guided Cufflinks[39] and StringTie[22] de novo assembly pipelines using the GRCm38 mouse or GRCh38 human genome, accessed through Ensembl[40]. Additionally, we used the novel method intron chain extension (ICE) for de novo detection of additional spliced transcripts, described below and in Supplementary Fig. 4.

For detection of spliced transcripts, ICE finds all splice junctions with a minimum threshold of supporting reads; a threshold of 10 reads per spliced intron was used in this study. For each splice junction, a nascent, two-exon transcript is created, and additional splice junctions are sought within a user-input maximum exon length in both genomic directions. If two splice junctions are located within the maximum exon length and share a similar number of supporting reads, the additional splice junction is merged into the nascent transcript, and the search for additional splice junctions continues. Once no additional splice junctions are found within a maximum exon length, transcript ends are determined based on read density at the genomic locus. Transcripts with identical intron chains are merged, and all overlapping transcripts are defined as splice isoforms of a single gene. For the transcriptome assembly reported here, de novo transcripts reported are limited to unannotated genome regions in Ensembl release 84[41] to avoid the potential misidentification of novel transcript variants of annotated genes. Using StringTie and ICE in tandem resulted in superior recapitulation of the human and mouse reference genomes compared to either tool alone (Supplementary Fig. 5b, c), thus the two transcriptome assemblies were merged. GTFs of the resulting expressed de novo genes for both mouse and human builds are given in the Supplementary Data 3 and 4.

To compare the output of these programs, we calculated a support score, $s$, for each spliced transcript generated by de novo assembly (Equation 1):

$$s = \frac{\min(r)}{\sum r} * i * \log_{10}\left(\sum r\right)$$

where $r$ is the number of intron-spanning reads for each intron in the transcript and $i$ is the number of introns in the transcript. A transcript score of zero indicates at least one intron with zero supporting intron-spanning reads. Scores for transcripts from each transcriptome assembly method are shown in Supplementary Fig. 5a.

**Gene expression analysis**. To quantify gene expression, HTseq-count[42] was to used count reads originating from the mouse and human reference transcriptomes (Ensembl release 84[41]) in addition to the genes identified from de novo assembly above. DESeq2 was used to normalize expression and perform differential expression analysis[43]. The ggplot2 R package and Prism (version 7.0c, GraphPad Software, Inc.) were used for data visualization[44].

Genes were considered expressed if an average of 20 or greater counts/gene/group was detected over the course of CD8$^+$ T cell differentiation. Of these, genes were considered differentially expressed if in any sample the absolute value of the log$_2$-fold gene expression change was >0.6 (fold change ~1.5) and the gene's $p$-value (Wald test) adjusted for multiple comparisons was <10$^{-3}$ compared to the naive condition. Annotated genes were divided into protein-coding and noncoding based on their Ensembl annotation. Novel genes were considered protein-coding if they were multi-exonic and had a protein-coding probability >0.8 as calculated by the Coding Potential Assessment Tool[23]; otherwise, they were considered noncoding.

Genes that were differentially expressed during CD8$^+$ T cell differentiation were clustered using leveraged affinity propagation, with the similarity matrix calculated using negative squared distances[24,45]. Twenty sweeps, using a randomly-selected 2% of all expressed genes, were performed.

PCA was performed with the FactoMineR package in R[46], using log-transformed expression values of all detected genes (or all detected genes of a given subclass, if applicable).

**Homology analysis**. Known human-mouse homolog pairs were retrieved from the Ensembl annotation[40] using the biomaRt package in R[47]. To discover potential human-mouse lncRNA homolog pairs, the mouse transcriptome (novel and reference genes) was masked using RepeatMasker[48] and aligned to the human transcriptome (novel and reference genes) using discontinuous megablast[49].To measure correlation between expression of homolog pairs between human and mouse effector and memory cells, Pearson's $r$ was calculated on the log$_2$-fold change of each type of cell compared to naive CD8$^+$ T cells.

## Data availability

Processed RNA-sequencing data is available in the Supplementary Data 1 and 2, and raw RNA-sequencing reads are available from the Sequence Read Archive under BioProject ID PRJNA412602. Custom computer code is available from the authors upon reasonable request. Data used to generate Figs. 1d and 4d are in the Source Data file, and processed data used to generate all other figures is available in the Supplementary Data.

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

## Acknowledgements

The authors thank Robert Karaffa, Sommer Durham, and the Emory University School of Medicine Flow Cytometry Core. W.H.H. is a Cancer Research Institute Irvington Fellow supported by the Cancer Research Institute. H.T.K. is supported by grant R00CA197804 from the National Cancer Institute of the National Institutes of Health.

## Author contributions

W.H.H. and J.G. performed experiments, W.H.H., N.P. and H.T.K. performed bioinformatics analyses, R. Akondy and D.J.M. provided assistance with data analysis and curation, R. Ahmed and H.T.K. provided funding and supervision and W.H.H. and H.T. K. wrote the manuscript with input from all authors.

## Additional information

**Competing interests:** The authors declare no competing interests.

