## [Peer Review File · Nature Communications]

Reviewers' comments:

Reviewer #1 (Remarks to the Author):

The authors have addressed most of the reviewer comments. Two additional comments:

1. In terms of the assessment of the quality of the transcriptome re-construction, the Supplementary Figure 5b-c should be split into two versions: one for coding genes and one for lncRNAs, since the focus of the manuscript is about lncRNAs which is more difficult to re-construct due to low abundance.
2. Also given the concerns of the accuracy of newly identified lncRNAs, the authors should show clear separations of different cell populations with only annotated lncRNAs or only multi-exonic lncRNAs. The new results showed that about 50% of monoexonic transcripts were questionable. It is unclear how this high error rate might impact the final results.

Reviewer #2 (Remarks to the Author):

The authors address most of my concerns and the manuscript is substantially improved. I recommend publication.

Response to reviewers for Hudson et al., Expression of novel long noncoding RNAs defines virus-specific effector and memory CD8⁺ T cells.

Authors' response is shown in bold. In addition to the changes listed below, we have made minor text edits for clarity and changed the presentation of Figures 1d and 4d to box-and-whiskers plots in accordance with *Nature Communications* policy. The underlying data and conclusions have not changed.

Reviewer #1 (Remarks to the Author):

The authors have addressed most of the reviewer comments. Two additional comments:

1. In terms of the assessment of the quality of the transcriptome re-construction, the Supplementary Figure 5b-c should be split into two versions: one for coding genes and one for lncRNAs, since the focus of the manuscript is about lncRNAs which is more difficult to re-construct due to low abundance.

This is a great suggestion. We have added a new figure (now Supplementary Figure 6) with coding and noncoding transcripts, for both mouse and human transcriptome assemblies.

2. Also given the concerns of the accuracy of newly identified lncRNAs, the authors should show clear separations of different cell populations with only annotated lncRNAs or only multi-exonic lncRNAs. The new results showed that about 50% of monoexonic transcripts were questionable. It is unclear how this high error rate might impact the final results.

This is another great suggestion. We have again added a new figure (now Supplementary Figure 8) showing separation of mouse cell types by PCA when using only multi- or single-exon lncRNA expression (panels a,b) or novel or annotated lncRNA expression (panels c,d). Separation patterns do not change based on noncoding transcript types used, but it is interesting that separation is clearer when novel transcripts are used. This is perhaps expected based on the higher dynamism of novel noncoding transcripts among CD8⁺ T cell subtypes compared to previously-annotated noncoding transcripts (see Figure 2c).

We thank the reviewer for his/her comments, which have greatly improved our manuscript.

Reviewer #2 (Remarks to the Author):

The authors address most of my concerns and the manuscript is substantially improved. I recommend publication.

We thank the reviewer for his/her comments.

REVIEWERS' COMMENTS:

Reviewer #1 (Remarks to the Author):

The authors addressed my concerns.

The figure legends should be checked before publication:

- 1) It is unclear what 'As in Supplementary Figure 6b-c,' means.
- 2) Supplementary Figure 8 is for mouse?

Reviewer #1 (Remarks to the Author):

The authors addressed my concerns.

The figure legends should be checked before publication:

- 1) It is unclear what 'As in Supplementary Figure 6b-c,' means.
- 2) Supplementary Figure 8 is for mouse

Reviewer #1 had two concerns regarding clarity of the legends for Supplementary Figures 6 and 8. We have edited these legends to address his/her concerns.